# Observation of momentum-band topology in PT-symmetric Floquet lattices

Shuaishuai Tong ⑩ , Qicheng Zhang ⑩ , Gaohan Li, Kun Zhang, Chun Xie & Chunyin Qiu ⑩ ✉

Momentum-band topology is a groundbreaking concept in multidisciplinary physics. Unlike the conventional energy-band topology, it defines a distinctive band topology within the energy Brillouin zone. Despite revolutionizing the paradigm of topological band theory, both theoretical and experimental studies of this new concept remain in their infancy. Here, we unveil the momentum-band topology in a PT-symmetric Floquet lattice, where the drive-induced momentum gap can be rigorously characterized by a quantized Berry phase. Experimentally, we synthesize the model using an acoustic cavity-tube structure coupled to custom-designed external circuits. By innovatively reconstructing the Floquet operator from measured time-domain wavefunctions, we extract the system's eigenstates and, for the first time, provide direct bulk evidence of momentum-band topology via momentum-band inversion and topological invariants. This is accompanied by a clear observation of time-localized interface states, thus providing a comprehensive examination of the temporal bulk-boundary correspondence. Our work paves the way for further experimental studies on the burgeoning momentum-gap physics.

Over the past decade, topological phases of matter have sparked widespread interest from condensed matter physics[1–3] to diverse classical wave systems[4–7]. In this context, bulk-boundary correspondence (see Fig. 1a) plays a central role by revealing the profound connection between the bulk energy-band topology and the existence of boundary states[1,8]. Mathematically, the bulk topology can be characterized by geometric invariants that capture the global property of bulk eigenstates across the momentum Brillouin zone (MBZ), such as the prominent Zak phase[9] and Chern number[10,11] for one- and two-dimensional systems. Furthermore, the introduction of periodic driving has led to the emergence of additional energy Brillouin zone (EBZ) and a series of non-equilibrium topological phases[12–20], including anomalous Floquet topological insulators[15,16] and topological space-time crystals with intertwined space-time symmetries[18–20]. Nevertheless, these studies primarily focus on the energy-band topology over the MBZ, with time modulation serving as an extra degree of freedom to control spatial topological interface modes (TIMs).

Momentum-band topology—first proposed in photonic time crystals[21], a type of dielectric media with refractive indices that undergo large, ultrafast periodic variations[21–26]—profoundly advances our understanding of temporal phases of matter. In this context, momentum gaps are formed by the interference of temporal refractions and reflections arising from time-periodic modulations[27–29], which showcase many intriguing properties, such as non-resonant lasers[25,30], subluminal Cherenkov radiation[31], and superluminal momentum gap solitons[32]. In particular, similar to conventional topological phases of matter, the global momentum-band topology in the EBZ results in temporally localized TIMs within momentum gaps[21,33–37] (Fig. 1b). Despite its revolutionary paradigm, the full theoretical elaboration of this emerging concept remains in its early stages. Furthermore, while experimental breakthroughs have been recently achieved in spatially discrete systems[34,35,37], the explorations are currently confined to the observation of TIMs, which provide only indirect evidence of momentum-band topology. Crucially, direct evidence of the bulk topology—the experimental characterization of the

Key Laboratory of Artificial Micro- and Nano-Structures of Ministry of Education and School of Physics and Technology, Wuhan University, Wuhan, China.
✉e-mail: cyqiu@whu.edu.cn

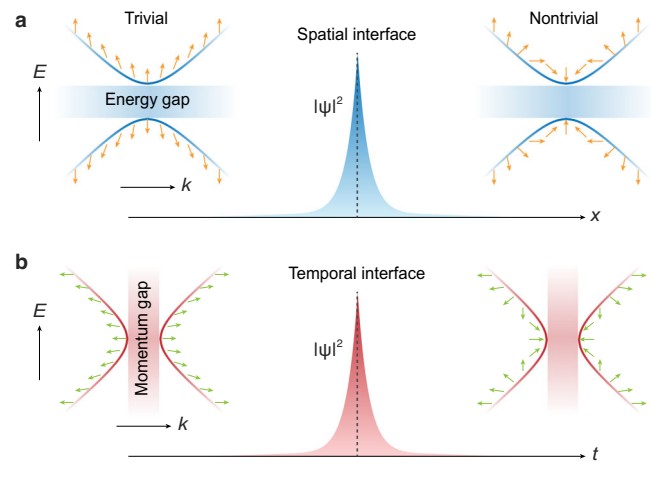

**Fig. 1 | Schematic of spatial and temporal bulk-boundary correspondence.**
**a** Energy-band topology enabling spatial topological interface modes (TIMs).
**b** Momentum-band topology enabling temporal TIMs. The evolutions of eigenstates, characterized by arrows, illustrate either trivial or nontrivial band topologies.

topological invariant itself—remains out of reach, hindered by the inherent complexity of time-varying systems and substantial technical challenges in extracting eigenstates. These critical bottlenecks underscore the pressing need for both theoretical transformation and experimental innovation.

In this work, we theoretically propose and experimentally demonstrate momentum-band topology in a parity-time (PT)-symmetric Floquet lattice. This model, which is broadly relevant to various experimental platforms, exhibits a nontrivial momentum gap induced by the periodic modulation of onsite gain and loss. The resulting momentum-band topology can be rigorously characterized by a quantized Berry phase defined over the EBZ. Experimentally, we emulate the Floquet lattice using an acoustic cavity-tube structure integrated with external circuits. The dynamic gain (loss) configuration is realized by positive (negative) feedback circuits, while the lattice momentum is synthesized by the phase shift of coupling circuits. Notably, by reconstructing the Floquet operator of our acoustic lattice, we extract both complex-valued energy spectra and Floquet eigenstates, and provide direct bulk evidence for the nontrivial momentum-band topology through the observation of band inversion and the characterization of bulk topological invariants. As complementary evidence of temporal bulk-boundary correspondence, we further observe time-domain TIMs within the Floquet momentum gap, which exhibit sound intensity localization at the nontrivial temporal interface. Our findings offer new insights into the novel momentum-gap phenomena and have fundamental implications for the broader field of nonequilibrium topological physics.

## Results
### Theoretical model
As illustrated in Fig. 2a, we consider a PT-symmetric Floquet lattice, where the two atoms in each unit cell are coupled via nearest-neighbor hopping $w$ and experience balanced gain and loss $\pm[\gamma_s + \gamma_d(t)]$. Here, $\pm\gamma_s$ denote static gain and loss, while $\pm\gamma_d(t) = \pm\gamma S(t)$ represent dynamic gain and loss, where $S(t) = \text{sgn}[\cos(2\pi\Omega t)]$ is a square-wave function with time period $T = 1/\Omega$. (For convenience, we use Hertz as the energy unit throughout this work.) The momentum-space Hamiltonian of this periodically driven lattice reads

$$\mathbf{H}(t) = \begin{bmatrix} i\gamma_s + i\gamma_d(t) & w + we^{ik} \\ w + we^{-ik} & -i\gamma_s - i\gamma_d(t) \end{bmatrix}. \quad (1)$$

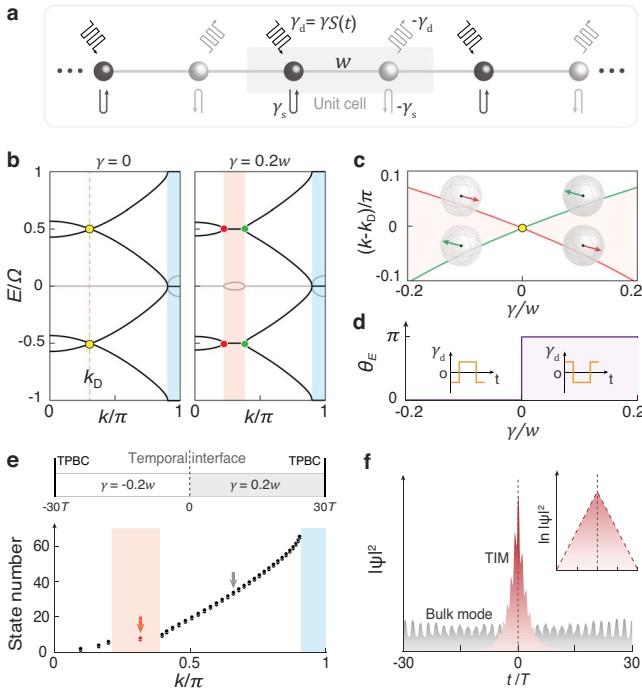

**Fig. 2 | Topological momentum gap in a PT-symmetric Floquet lattice. a** Tight-binding model. It features nearest neighboring hopping $w$, static gain/loss $\pm\gamma_s$, and dynamic gain/loss $\pm\gamma_d(t) = \pm\gamma S(t)$, with $S(t)$ being a $T$-period square wave function. **b** Band structures for the lattices without (left) and with (right) dynamic gain/loss applied, where the dark and grey lines represent the real and imaginary parts of the energy, respectively. The nonzero dynamic gain/loss splits the Floquet Dirac points (yellow dots) into a pair of Floquet EPs (red and green dots). The orange and blue shaded areas label the Floquet and static momentum gaps, respectively. **c** $\gamma$-dependence of the Floquet-EP momenta, indicating a band inversion in momentum. Insets: typical Floquet-EP states (arrows) sketched on the Bloch sphere.
**d** Berry phase $\theta_E$ plotted as a function of $\gamma$, where the insets illustrate the profiles of dynamic gain/loss for $\gamma < 0$ and $\gamma > 0$. **e** Momentum spectrum (lower panel) of a temporal domain-wall structure (upper panel) with a temporal periodic boundary condition (TPBC) and a spatial Bloch boundary condition. **f** Probability density distributions of the bulk mode (gray) and TIM (red) across the domain-wall structure. The latter is well captured by the simplified low-energy model around the Floquet Dirac point (red dashed line, inset).

The quasi-energy spectrum can be solved from the effective Hamiltonian $\mathbf{H}_{eff} = (i2\pi T)^{-1}\ln\mathbf{U}$, where the Floquet operator

$$\mathbf{U} = e^{-i2\pi\mathcal{T}\int_0^T \mathbf{H}(t)dt}$$

with $\mathcal{T}$ being time-ordering operator. Without loss of generality, we consider $\Omega = 3.5w$, $\gamma_s = 0.3w$, and $\gamma = 0.2w$. Figure 2b presents the band structure for the system (right panel), compared to the case without dynamic modulation (left panel). For clarity, we provide the quasi-energy bands within two EBZs and half of the MBZ [notice that $E(-k) = E(k)$]. Clearly, it shows that the Floquet Dirac points at the EBZ boundaries, which are crossed by neighboring quasi-energy bands at $\gamma = 0$ because of the virtual Floquet band folding, split into pairwise Floquet exceptional points (EPs) and form Floquet momentum gaps (featuring pairwise growing and decaying modes with complex-valued quasi-energies) due to the non-zero dynamic modulation, known as Floquet PT-symmetry breaking[38–42]. Note that the momentum gap around $k = \pi$, referred to as the static momentum gap, is induced by a non-zero $\gamma_s$. The introduction of the static momentum gap serves two primary purposes: it will facilitate the characterization of topological invariants in the EBZ and enable a direct comparison to the Floquet momentum gap with nontrivial topology. (This

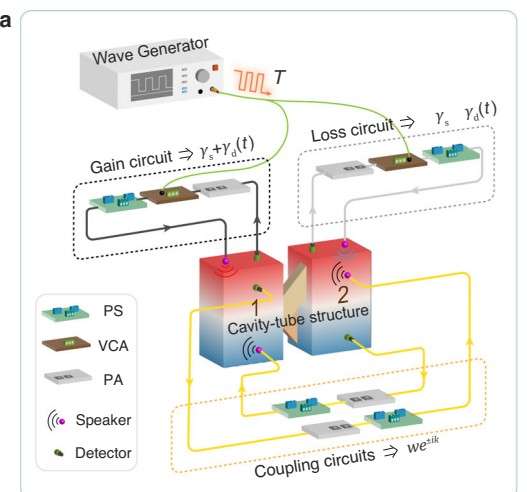
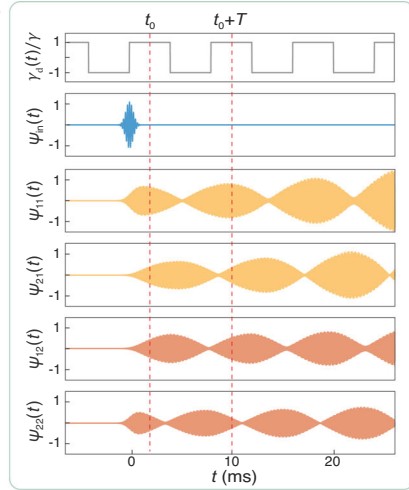

**Fig. 3 | Acoustic emulation of the PT-symmetric Floquet lattice in synthetic space. a** Schematic diagram of our experimental setup. The acoustic cavities mimic the atoms, and the narrow tubes between them emulate the reciprocal intracell hopping, respectively. A gain circuit, composed of a detector, a preamplifier (PA), a voltage-controlled-amplifier (VCA), a phase shifter (PS), and a speaker, introduces the onsite gain by providing positive feedback to cavity 1. Similarly, a loss circuit introduces the on-site loss by providing negative feedback to cavity 2. The temporal modulation is synchronized by a $T$-period square-wave voltage generated by a wave generator. A couple of phase-controllable coupling circuits enable the intercell hopping $we^{ik}$. **b** Measured time-domain wavefunctions $\mathbf{\psi}_1(t) = (\psi_{11}, \psi_{21})^T$ and $\mathbf{\psi}_2(t) = (\psi_{12}, \psi_{22})^T$, which are generated by injecting a Gaussian sound signal $\psi_{in}(t)$ into cavities 1 and 2, respectively. The sound signals at $t_0$ and $t_0 + T$ are extracted for reconstructing the effective Hamiltonian. The experimental data, exemplified by $k = 0.3\pi$, correspond to the system with $w = 36$ Hz, $\gamma_s = 11$ Hz, $\gamma = 7$ Hz, and $\Omega = 125$ Hz.

comparison highlights the unique momentum-gap topology inherent to our PT-symmetric Floquet lattice.)

Now, beyond previous studies on PT-symmetry breaking[38–42], we investigate the topology of the momentum gap induced by the dynamic modulation of gain and loss. To do this, we first examine the evolution of the Floquet-EP states as a function of the parameter $\gamma$, which characterizes the dynamic gain and loss in our PT-symmetric Floquet lattices. As shown in Fig. 2c, the $\gamma$-dependence of the Floquet-EP momenta reveals that the two EP states swap their momentum order when crossing the Floquet Dirac point at $\gamma = 0$. Similar to the energy-band inversion in conventional topological band theory[1,43], the momentum-band inversion marks a signature of topological transition. In our PT-symmetric Floquet system, the momentum-band topology can be characterized by a quantized Berry phase defined in the EBZ

$$\theta_E = i\oint_{\text{EBZ}} \psi^\dagger(E)\partial_E \psi(E)dE, \tag{2}$$

with $\psi(E)$ being the eigenstate of the effective Hamiltonian $\mathbf{H}_{\text{eff}}$. Figure 2d presents the $\gamma$-dependent Berry phase calculated for the momentum band between the static and Floquet momentum gaps. It shows two topologically distinct phases: $\theta_E = 0$ for $\gamma < 0$ and $\theta_E = \pi$ for $\gamma > 0$. According to the bulk-boundary correspondence, one can expect temporally localized states at a time interface formed by the two phases. To demonstrate this, as shown in Fig. 2e, we construct a temporal domain-wall structure composed of the Floquet lattices with $\gamma = \pm 0.2w$, and calculate the momentum spectrum under a time-periodic boundary condition and spatial Bloch boundary condition (see Supplementary Note 3 for details). The spectrum reveals isolated topological states (red dot) within the Floquet momentum gap. (Note that owing to the implementation of the time-periodic boundary condition, all eigenstates have zero imaginary components in their eigenenergy, and the in-gap growing and decaying modes are excluded from the momentum spectrum). Accordingly, the time-domain probability density distributions $|\psi|^2$ in Fig. 2f demonstrate that the in-gap TIM (red area) is strongly localized around the time

interface, in sharp contrast to the temporally extended bulk state (grey area). (Note that the probability density distribution of the states is spatially uniform due to the implementation of spatial Bloch boundary condition.) Interestingly, the exponential decay of the TIM can be precisely captured by a low-energy model with a momentum operator $M_D(\delta E) = v_D^{-1}(m\sigma_y - \delta E\sigma_z)$ — a momentum version of the effective Dirac Hamiltonian. Here $\delta E$ represents the energy deviation from the Floquet Dirac point, $v_D$ is Dirac velocity, and $\sigma_{y,z}$ are Pauli matrices, respectively. Resembling the conventional Dirac model that characterizes the energy-gap topology, the momentum-gap topology depends on the sign of Dirac mass $m$, which is proportional to $\gamma$. As the consequence, an interface separating the time domains with Dirac masses of opposite signs supports nontrivial temporal TIMs. (see Supplementary Note 1 for details of the low-energy model).

### Acoustic implementation of PT-symmetric Floquet lattices

Experimentally, we simulate the PT-symmetric Floquet lattice utilizing a passive acoustic cavity-tube structure[44–46] supplied with external circuits[47–50]. As sketched in Fig. 3a, the air-filled cavities emulate the lattice sites with onsite energy $f_0 = 5100$ Hz, while the narrow tubes between them provide static and reciprocal intracell coupling $w = 36$ Hz. To implement the desired gain (loss) configuration, we utilize a set of time-dependent positive feedback circuit composed of detectors, speakers and other control units to provide the gain $\gamma_s + \gamma_d(t)$ in cavity 1, while using a set of negative feedback circuits to introduce the balanced loss $-\gamma_s - \gamma_d(t)$ in cavity 2. The effective onsite gain (loss) is introduced when the feedback sound pressure from the speaker is in-phase (out-of-phase) with respect to the sound at the top of the cavity. The temporal modulation is synchronized by a square-wave voltage generated by a waveform generator. In addition, two sets of unidirectional coupling circuits are introduced to mimic the intercell hoppings $we^{\pm ik}$[51–53], where the lattice momentum $k$ is mapped to the phase change of phase shifters. More details about the experimental setup are provided in the Methods. Ultimately, we synthesize the PT-symmetric Floquet lattice governed by the $k$-dependent Hamiltonian in Eq. (1). Note

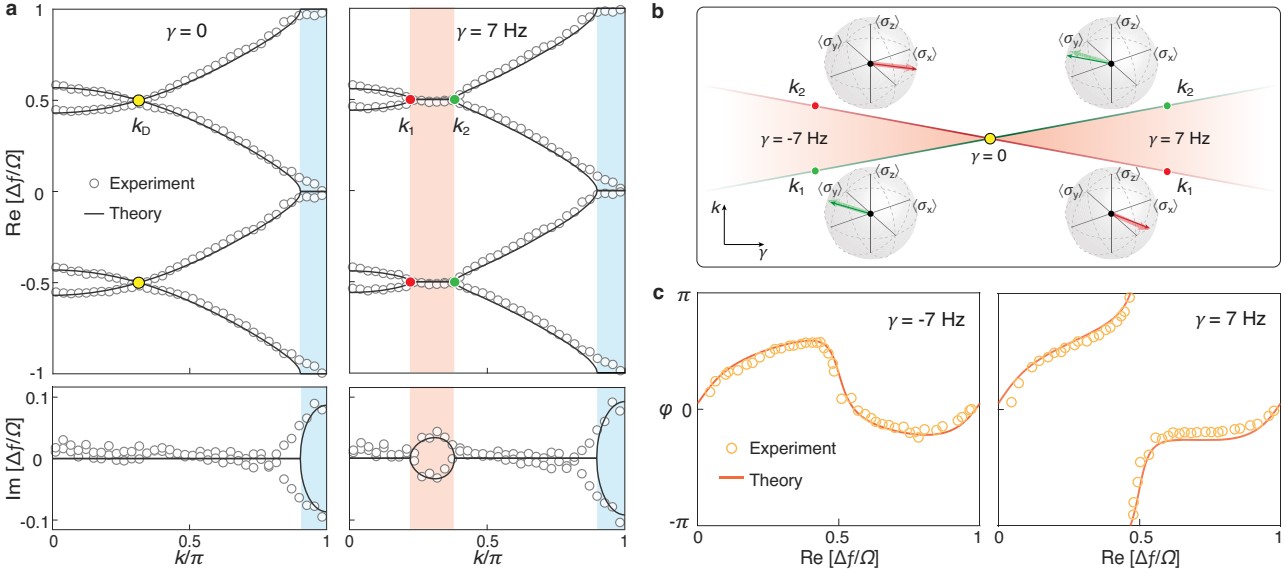

**Fig. 4 | Experimental evidence for momentum-band topology in the EBZ.**
**a** Comparative energy-momentum spectra measured for the systems with $\gamma = 0$ (left) and $\gamma = 7$ Hz (right), respectively. For clarity, here the real and imaginary spectra are plotted separately, and the frequency $\Delta f$ is measured from the resonant frequency of a single cavity. The blue shaded area labels the static momentum gap induced by the static gain/loss, while the orange shaded area highlights the Floquet momentum gap driven by the dynamic modulation. **b** Floquet-EP states (thick arrows) measured for the systems with $\gamma = -7$ Hz (left) and $\gamma = 7$ Hz (right), compared with the simulation results (thin arrows). **c** Phase evolutions of PT-eigenvalues within the EBZ for the systems of $\gamma = -7$ Hz and $\gamma = 7$ Hz, demonstrating the trivial and nontrivial momentum-band topologies, respectively. The experimental data (circles) agree well with theoretical results (solid lines).

that our experimental setup allows flexible momentum selection by simply regulating the phases of coupling circuits. This facilitates a precise measurement of the energy-momentum spectrum and eigenstates, thereby enabling a direct characterization of the momentum-band topology in the EBZ.

Here, to experimentally extract the complex-valued energy-momentum spectra as well as temporal eigenstates of our acoustic systems, we propose a highly practical approach to reconstruct the effective Hamiltonian $\mathbf{H}_{\text{eff}}$. As shown in Fig. 3b, we inject a pulsed acoustic signal $\psi_{\text{in}}(t)$ into cavity $i$, and measure the resultant sound response in cavity $j$, denoted by $\psi_{ji}(t)$ with $i,j = 1, 2$. This gives two linearly independent wavefunctions $\boldsymbol{\psi}_1(t) = (\psi_{11}, \psi_{21})^{\text{T}}$ and $\boldsymbol{\psi}_2(t) = (\psi_{12}, \psi_{22})^{\text{T}}$. Theoretically, for the wavefunction $\boldsymbol{\psi}_i(t)$ that evolves freely over time, $\boldsymbol{\psi}_i(t_0)$ and $\boldsymbol{\psi}_i(t_0 + T)$ are related by the Floquet operator, $\boldsymbol{\psi}_i(t_0 + T) = \mathbf{U}\boldsymbol{\psi}_i(t_0)$. Consequently, using the two experimentally measured states, we reconstruct the $2 \times 2$ Floquet operator via

$$\mathbf{U} = \left[ \boldsymbol{\psi}_1(t_0 + T), \boldsymbol{\psi}_2(t_0 + T) \right] \left[ \boldsymbol{\psi}_1(t_0), \boldsymbol{\psi}_2(t_0) \right]^{-1}, \quad (3)$$

which further gives the effective Hamiltonian of the Floquet lattice: $\mathbf{H}_{\text{eff}} = (i2\pi T)^{-1}\ln\mathbf{U}$. More specifically, as shown in Fig. 3b, we consider a short-duration Gaussian wavepacket $\psi_{\text{in}}(t) = \exp(-t^2/\Delta t^2)\exp(i2\pi f_0 t)$ with $\Delta t = 0.5$ms, and set $t_0 = 2$ms to ensure sufficient temporal separation between the resultant sound responses over a single $T$-period and the incident pulse signal, since $\psi_{\text{in}}(t) \to 0$ when $t > t_0$. Based on the reconstructed effective Hamiltonian $\mathbf{H}_{\text{eff}}(k)$, we obtain the quasi-energies and eigenstates of the PT-symmetric Floquet lattice for each momentum, as well as the energy-momentum spectrum by sweeping the phase of coupling circuits over $2\pi$. Our experimental scheme, based on the reconstructed Floquet operator, enables direct measurement of complex-valued energy spectra in non-Hermitian Floquet lattices. This technique significantly differs from previous ones, where only real-valued band structures are extracted from space-time Fourier spectra[54–56].

## Experimental evidence for momentum-band topology

Figure 4a presents the complex energy-momentum spectrum measured for the acoustic system with $\gamma = 7$ Hz, in comparison with that of the static case ($\gamma = 0$). It shows that while both systems exhibit a static momentum gap around $k = \pi$, the presence of dynamic gain and loss opens a momentum gap at the Floquet Dirac point near $k = 0.3\pi$, accompanied by complex-valued energies between the Floquet-EP momenta $k_1 \simeq 0.21\pi$ and $k_2 \simeq 0.39\pi$. (The deviation between the experimental and theoretical results arises from noise and imperfections in system parameters[49], e.g., the hopping $w$, the gain and loss parameters $\gamma_s$ and $\gamma$.) The presence of the Floquet momentum gap is further corroborated by the observation of exponentially growing sound intensity over time, as detailed in the Supplementary Note 4.

Below, we directly demonstrate the momentum-band topology in the EBZ by analyzing the eigenstates of our experimentally reconstructed effective Hamiltonian. Figure 4b shows the measured Floquet-EP states for the systems with $\gamma = \pm 7$ Hz, indicated by thick arrows on the Bloch spheres, which match well the theoretical predictions (thin arrows). [Note that the experimental energy-momentum spectrum of $\gamma = -7$ Hz, not presented in Fig. 4a, is nearly indistinguishable from that of $\gamma = 7$ Hz.] As predicted, the two Floquet-EP states exchange their momenta ($k_1$ and $k_2$) when the sign of $\gamma$ is reversed—an essential signature of the momentum-band inversion in our acoustic Floquet lattice. Furthermore, we characterize the Berry phase based on the experimentally measured eigenstates. Notably, due to the PT-symmetric nature of the real-energy eigenstates in between the static and Floquet momentum gaps, the Berry phase can be equivalently expressed through the winding angle of the PT eigenvalue $e^{i\varphi(E)} = \psi^\dagger(E)\mathbf{PT}\psi(E)$ around the origin of the complex plane (see Supplementary Note 2 for details). This results in an alternative formulation for the Berry phase

$$\theta_E = \frac{1}{2}\oint_{\text{EBZ}} \partial_E \varphi(E) dE. \quad (4)$$

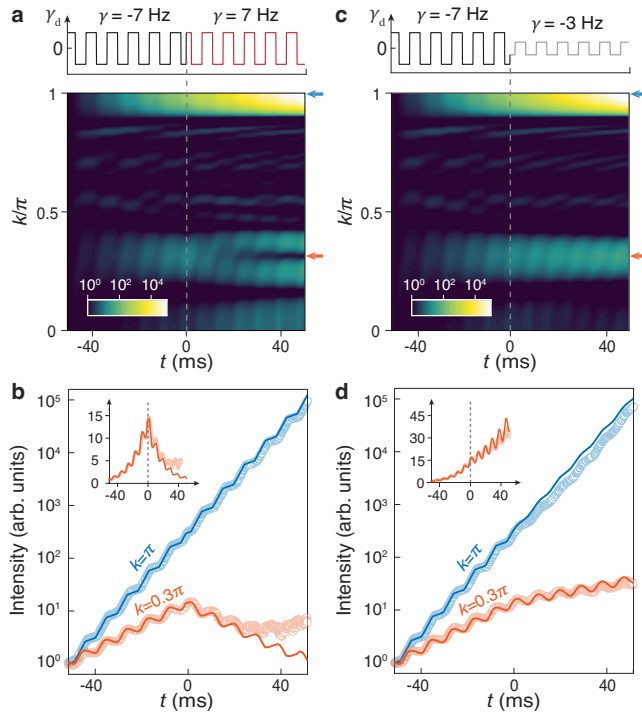

**Fig. 5 | Experimental observation of acoustic temporal TIMs. a** Momentum-resolved time evolution of sound intensity simulated for a nontrivial temporal interface system. The orange and blue arrows mark the momenta $k = 0.3\pi$ and $k = \pi$, respectively. **b** Sound intensities at these two momenta, with experimental data (circles) closely matching the simulations (lines). A base−10 logarithmic scale is used to better visualize the exponential behavior. The data for $k = 0.3\pi$, along with their linear-scale plot (inset), clearly show the emergence of a TIM at the temporal interface ($t = 0$ ms). **c, d** Similar to **a** and **b**, but for a trivial temporal interface system, showing no TIM at the interface.

This expression greatly simplifies our experimental characterization. Figure 4c shows the evolution of measured phase angles $\varphi(E)$ for the systems with $\gamma = \pm 7$ Hz. For the system with $\gamma = -7$ Hz, the phase exhibits no winding across the EBZ, indicating a trivial topology with $\theta_E = 0$. In contrast, for $\gamma = 7$ Hz, the phase winds by $2\pi$, identifying the nontrivial topology with $\theta_E = \pi$. All experimental results (circles) agree excellently with the theoretical predictions (solid lines), providing clear evidence distinguishing the momentum-band topologies of the two systems in the EBZ.

### Observation of acoustic temporal TIM

To demonstrate the intriguing temporal TIM in our acoustic system, we investigate the wave dynamics around the temporal interface at $t = 0$, realized by setting $\gamma = -7$ Hz for $t < 0$ and $\gamma = 7$ Hz for $t > 0$. Figure 5a presents the momentum-resolved sound intensity simulated as a function of time, with an initial pulse excitation applied to cavity 1. In contrast to the monotonic signal growth within the static momentum-gap around $k = \pi$, the sound intensity within the Floquet momentum-gap (near $k = 0.3\pi$) manifests a fine structure. Specifically, within an extremely narrow momentum range, the sound signal initially increases but then decreases after the temporal interface—a hallmark characteristic of temporal TIM. To experimentally identify these fundamental signatures, we focus on the momenta $k = 0.3\pi$ and $k = \pi$, and switch the gain and loss configuration to form a nontrivial temporal interface by simply controlling the voltage-controlled-amplifier via the waveform generator. Compared to the case of $k = \pi$, which manifests a stable exponential growth of sound signal, the measured sound intensity for $k = 0.3\pi$ clearly confirms the emergence of the

temporally localized TIM (Fig. 5b). The deviation between the experimental and simulated results for $t > 25$ ms is primarily attributable to imperfections in system parameter settings[49]. Note that the intensity pattern around the interface state is highly sensitive to the momentum (see Supplementary Note 5), since the Floquet lattices at both sides of the time interface support temporally growing in-gap modes, in addition to decaying modes akin to those in conventional spatial topological insulators[21,34]. Furthermore, we present the comparative intensity spectra for a temporal interface composed of two topologically trivial systems with $\gamma = -7$ Hz and $\gamma = -3$ Hz (Fig. 5c, d). As expected, the intensities in both momentum gaps increase monotonically with time and show no temporal TIMs in this case.

## Discussion

We theoretically unveil the momentum-band topology in Floquet PT-symmetric lattices and elucidate its intrinsic connection with fundamental PT-symmetric physics. Built upon a widely applicable tight-binding model, our findings exhibit broad interdisciplinary relevance, spanning condensed matter physics, integrated photonics, electric circuits, and cold-atom systems. Furthermore, we present a comprehensive, wavefunction-level experimental study of the momentum-band topology using acoustic Floquet lattices. We not only provide the first direct bulk evidence of momentum-band topology via band-inversion and topological invariants, but also demonstrate the intriguing TIM in real physical time, thus complementarily highlighting the role of temporal bulk-boundary correspondence.

Looking ahead, the conclusive identification of momentum-band topology opens exciting directions for further experimental exploration, such as temporal Thouless pumping in lattice systems. The inherent causality constraint in temporal systems may give rise to phenomena fundamentally distinct from their spatial counterparts—for instance, the directionality arising from the one-way nature of time and the breakdown of quantization caused by non-Hermiticity. In addition, the key experimental technique developed here—reconstructing the Floquet operator via measuring the temporal evolution of wavefunctions—effectively addresses the long-standing challenge of probing the complex-valued quasi-energy spectrum and eigenstates of non-Hermitian Floquet systems. This method holds broad applicability for future studies of Floquet non-equilibrium systems, enabling comprehensive, wavefunction-level experimental access to the dynamics of periodically driven systems. Last but not least, our theoretical and experimental results demonstrate significant potential for exploring a wide range of temporal matter physics in dynamic PT-symmetric systems through the control of time-dependent gain and loss. This includes, for example, temporal quasicrystals[57], time-domain Anderson effects[58,59], and temporal moiré superlattices[60,61]. These innovative concepts could enable the design of functional devices with capabilities that go beyond conventional Floquet systems.

## Methods

### Experimental setup

Experimentally, we simulate the PT-symmetric Floquet lattice utilizing an acoustic cavity-tube structure equipped with external circuits. The acoustic cavity-tube structure consists of two identical air cavities connected by two cross-linked narrow tubes. Each cavity has a dimension of $26.7 \times 20.0 \times 33.6$ mm$^3$. The narrow tubes, tilt at an angle of 40 degrees with respect to the third dimension, have a length 19.2 mm and a cross-section area $4.5 \times 4.5$ mm$^2$. The acoustic sample is 3D-printed by photosensitive resin at a thickness of 5.0 mm. We consider the dipole mode in the individual cavity, which has a resonant frequency $f_0 = 5100$ Hz at sound speed 343 ms$^{-1}$. When coupled by the narrow tubes, the cavity modes split into an antisymmetric mode at 5064 Hz and a symmetry mode at 5136 Hz, which give rise to a reciprocal acoustic coupling of $w = 36$ Hz.

In addition to the passive cavity-tube structure, external circuits composed of detectors, speakers and other control units are employed to realize the desired gain (loss) configurations and reciprocal intercell couplings. More specifically, as shown in Fig. 3a, the gain circuit that connects with cavity 1 provides the dynamic gain $\gamma_s + \gamma_d(t)$. The workflow can be introduced as follows: the sound signal at the top of cavity 1 is picked up by a detector, modulated by a preamplifier, a voltage-controlled-amplifier (VCA), a phase shifter, and finally feedbacked to cavity 1 through a speaker. To mimic the onsite gain, the emission from the speaker is in-phase with respect to the sound at the top of cavity 1[48]. The gain amplitude, linearly related to the VCA's amplification factor, alternates in time between $\gamma_1 = 4\,\text{Hz}$ and $\gamma_2 = 18\,\text{Hz}$ according to a square wave voltage created by a wave generator. (Note that the voltage output from the wave generator does not directly participate in the energy exchange of the gain (loss) circuits, but rather dynamically switches the amplification factor of the VCA, thereby realizing the time-dependent amplitude modulation of gain and loss. The periodic modulation of gain and loss does not result in a simple cycle of sound energy injection and cancellation, see, e.g., Fig. 5c, d. This behavior reflects the dynamic nature of the underlying physical parameters.) This setup realizes a static gain $\gamma_s = \frac{1}{2}(\gamma_1 + \gamma_2) = 11\,\text{Hz}$ and a dynamic gain $\gamma = \frac{1}{2}(\gamma_2 - \gamma_1) = 7\,\text{Hz}$. The loss circuit connected to cavity 2 is nearly identical to the gain circuit, while the emission from the speaker is out-of-phase with respect to the sound at the top of cavity 2 (realized by adjusting the phase shifter). This enables a time-dependent loss with $\gamma_1 = -4\,\text{Hz}$ and $\gamma_2 = -18\,\text{Hz}$, or equivalently $\gamma_s = -11\,\text{Hz}$ and $\gamma = -7\,\text{Hz}$.

To synthesize a periodic lattice using a double-cavity system, we implement the momentum-dependent intercell couplings $we^{\pm ik}$ using a pair of unidirectional coupling circuits, which connect the two cavities in opposite directions. More specifically, the sound signal in cavity 1 (2) is picked up by a detector, then modulated by a preamplifier, a phase shifter, and finally output into cavity 2 (1) via a speaker. Note that the hopping strength ($w$) is roughly proportional to the amplification factor ($G$) of the preamplifier, while the phase ($\pm k$) of the unidirectional coupling is nearly linear with the phase delay ($\theta$) introduced by the phase shifter, respectively[49]. This setup allows us to generate any desired $k$-dependent complex coupling $we^{\pm ik}$ by regulating $G$ and $\theta$ together. To experimentally reproduce the band structure, we vary $\theta$ from 0° to 180° at a step of 5°, which emulates the momentum $k$ ranging from 0 to $\pi$. Ultimately, we realize the momentum-dependent lattice model presented in Eq. (1). Unlike conventional finite-lattice approaches, which require carefully engineered initial wave packets for momentum selection, our experimental setup enables flexible momentum control by directly adjusting the phases of the coupling circuits. This facilitates a precise measurement of the energy-momentum spectrum and eigenstates, thereby providing a direct demonstration of the band topology in EBZ. Note that the presence of uniform background loss $\gamma_0$ does not affect the essential physics but introduces an extra exponential decay $e^{2\pi\gamma_0 t}$ in the time-domain wavefunction. In this work, this decay is compensated by multiplying an exponential factor $e^{-2\pi\gamma_0 t}$ to the measured wavefunction (see Supplementary Note 6). However, to ensure a sufficiently long detectable sound signal, we use two static in-phase feedback circuits (not shown in Fig. 3a) to reduce the background loss in both cavities to $\gamma_0 = -13\,\text{Hz}$.

## Data availability

All the data supporting this study are available in the paper and Supplementary Information. Additional data related to this paper are available from the corresponding authors upon request.

## Code availability

The custom codes for this study that support the findings are available from the corresponding authors upon request.

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

## Acknowledgements

This project is supported by the National Key Research and Development Program of China under Grant No. 2023YFA1406900 (C.Q.), the National Natural Science Foundation of China under Grants No. 12374418 (C.Q.), No. 12304495 (S.T.) and No. 12104346 (Q.Z), the China Postdoctoral Science Foundation under Grant No. 2024M762462 (S.T.) and the Fundamental Research Funds for the Central Universities (C.Q.).

## Author contributions

C.Q. initiated and supervised the project. S.T. developed the theory and carried out numerical simulations. S.T. built the experimental setup and performed the experiments with contribution from Q.Z. G.L. K.Z. and C. X. S.T. and C.Q. analyzed the data and co-wrote the draft. All authors participated in discussions.

## Competing interests

The authors declare no competing interests.
