## [Transparent Peer Review file · Nature Communications]

Observation of momentum-band topology in PT-symmetric Floquet lattices

Corresponding Author: Professor Chunyin Qiu

Version 0:

Reviewer comments:

Reviewer #1

(Remarks to the Author)
Referee report.

The authors present an experimental demonstration of extracting the topological invariant of the momentum bandgap in time-modulated acoustic system. Momentum band-gaps occur in time-periodic systems in which the eigen-states form continuous bands that are gapped in the momentum. Unlike the more known energy bandgaps, here “forbidden” states (states in the gaps) do not conserve energy (draw or deplete energy from/to the modulation. Momentum band-gaps are relatively hard to realize (compared with energy gaps) especially in a direct way. As part of the increase in time-modulated system in recent years, several momentum band-gap phenomena were demonstrated, including the interface states in time that were also demonstrated here. Additionally, band topology is a well studied concept in energy band-gaps, but not so much for momentum band-gaps and were only proposed recently. Therefore, the observation presented here, of the topological invariant of a momentum gap is a fundamental result and the first direct and convincing experimental verification of the underlying bulk topology, and an important milestone and the research of time-dependent system. Additionally, the results are convincing and unambiguous, and the writing is pretty good. The additional result on the connection with the edge-states and non-Hermitian physics makes the work even richer. Therefore, I believe this manuscript is suitable to be published in Nature communications after minor revisions.

1. in Fig.5b, regarding fact that the sound grows (and deviates from theory). This seems like the behavior seen in other works, in which the amplified mode that is the pair mode of the decaying mode that creates the edge-state start growing. If that is the case, I think the explanation can talk a little more about it – since this effect is seen in other works which showed only the edge-states (theory and experiment).
2. in Fig.2c the arrows in the sphere are too small and it's hard to understand it or see it. It would be good to make it clearer.
3. Fig.2e – no x label
4. Methods Details on Loss Compensation: In the Methods section, the authors mention a uniform background loss γ_0 and state that this can be compensated for. Could they clarify if the data presented in the main figures (e.g., Fig. 5) has been compensated? If so, briefly describing the procedure and perhaps showing a raw vs. compensated data plot in the Supplementary Information would enhance the reproducibility of the work.
5. what is the meaning of virtual in virtual Dirac point? It is not clear to me. Maybe the authors can put a sentence on that.
6. Outlook on Causality in Temporal Pumping: The discussion of future work mentions that causality may lead to fundamentally distinct conclusions for temporal Thouless pumping. This is a very intriguing statement. Adding a single sentence to briefly speculate on what kind of new physics might be expected (e.g., directionality, quantization breakdown) would make the outlook even more exciting for the reader.

Reviewer #2

(Remarks to the Author)

In the manuscript titled by “Observation of momentum-band topology in acoustic PT-symmetric Floquet lattices”, authors focus on momentum band topology in the acoustic cavities. The static model is a PT symmetric two cavities which open $k=\pi$ momentum gap under PBC. Then, they added on-site time varying amplitude term in the type of alternative gain and loss, and the time-varying functions of two cavities are opposite to each one. They found that another momentum gap is opened away from $k=\pi$ which can support a temporal analogy of Jackiw rabbi mode- localized in a temporal interface. The

experiment is based on two acoustic helmholtz resonator coupled by two tubes which is not tunable. The periodic boundary condition is realized by two phase shifters via electric circuit, and gain and loss are also given by electric circuit which act as additional in-phase and out-phase amplitude injection respectively.

Overall, the novelty of this paper is on the measurement of this acoustic 1D Floquet bands' winding number and states. However, considering the slim novelty and the following concerns, I cannot recommend its publication in Nat. com at this stage.

1 Clarification Required on Gap Opening:

Does the opening of the k -gap (not $k=\pi$ one) require the static gain and loss? The temporal interface mode is attributed to Floquet SSH temporal domain model, as an analogy of Jackiw-Rebbi zero modes (from their formula). The static gain and loss gives $k=\pi$ gap, which exhibits large amplitude exponential growth factor.

2 Concerns Regarding "Time Modulation" Experimental Implementation:

It is hard to say the time varying in this manuscript is genuine time modulation. The mechanism of this experiment is based on fast electric sensor for acoustic amplitude, fast operation (compare with acoustic frequency), and injection (the ending of this loop). It is a time-periodic energy cancellation or injection, instead of modulating some physical parameters. On this basis, it makes little sense for "a broad range of temporal matter physics. This includes, for example, temporal quasicrystals, time-domain Anderson effects, and temporal Moiré superlattices".

3 Ambiguities in Temporal Interface State Observations (Fig. 5 and Fig. S4):

The temporal localization is only at a specified k at the k -gap around $0.2-0.5\pi$. Yet, a small deviation from that value makes the intensity exponentially grow. How such exponentially growing modes exist? From the spectrum, it should be evanescent one as in the k gap there should be only interface states at the specified k realized via phase shifters in circuits. For such acoustic experiment with fast electric device control, such deviation should be clarified. Also, the increasing of the tail seems not localized when observing it beyond 40ms.

4 Insufficient Resolution near Exceptional Points (EP)

In the other EP experimental observations, the relative low resolution around EP is mostly due to the parameter shift or disorder. Yet for the experiment here, as all the couplings, periodic boundary condition k , time varying, gain and loss in acoustic cavities are precisely controlled by electric device- which is quite accurate and fast enough for varying the acoustic amplitude, acoustic amplitude detection & injections. Why there is still low resolution around EP in the results shown in Fig. 4?

5 Insufficient Literature Coverage:

For example, The model without static gain and loss has been revealed and experimentally checked in "Liu, Weijie, et al. "Floquet parity-time symmetry in integrated photonics." Nature Communications 15.1 (2024): 946.". In addition, It is nice idea to use two different input to rebuild the Floquet operator. This is a kind of tomographic of Floquet operator. Yet, the prior literature is not properly mentioned. I recommend author refer to Asapanna, Rajesh, et al. "Observation of extrinsic topological phases in Floquet photonic lattices." Physical Review Letters 134.25 (2025): 256603." Also, for the realization of periodic boundary condition, it has been proposed and realized experimentally in both acoustic and microwave platforms. Please refer to: Chen, Zhao-Xian, et al. "Direct measurement of topological invariants through temporal adiabatic evolution of bulk states in the synthetic Brillouin zone." Physical Review Letters 134.13 (2025): 136601. Zhang, Zhe, Pierre Delplace, and Romain Fleury. "Anomalous topological waves in strongly amorphous scattering networks." Science Advances 9.12 (2023): eadg3186.

6 Incomplete Information for Fig. 2e and Fig. 2f:

The real-space configuration for the state depicted in Fig. 2f and the imaginary component in Fig. 2e are not described. This information is crucial for proper interpretation and understanding of the mode structure.

7 Clarification of "Virtual Dirac Point":

The concept and implications of a "virtual Dirac point" introduced on page 5 (first line) remain unclear. This discussion must be refined for better readability and scientific clarity.

8 Figure Improvement:

The azimuthal axis in Fig. 2e should be explicitly labeled for better clarity.

Reviewer #3

(Remarks to the Author)

This manuscript presented a comprehensive experimental study of momentum-band topology in a PT-symmetric Floquet lattice, addressing the lack of direct bulk evidence for this emerging topological concept. Using an acoustic cavity-tube system coupled with active feedback circuits, the researchers synthesized a Floquet lattice where periodic gain/loss modulations induced momentum gaps. By reconstructing the effective Hamiltonian from time-evolved wavefunctions, they experimentally confirmed momentum-band inversion and quantified the topological invariant in the energy Brillouin zone, providing the evidence of momentum-band topology. Additionally, they observed temporally localized interface states at a topological domain wall, verifying the temporal bulk-boundary correspondence. The manuscript presents an interesting experimental exploration of momentum-band topology in acoustic systems, and it is well-written. However, I still have a few more questions and suggestions:

1. Please check whether the x-axis label is missing in Fig. 2e and Fig. 4b.
2. The manuscript states that the narrow tube provides a static and reciprocal intracell coupling ($\omega=36$ Hz), while also introducing two sets of unidirectional coupling circuits to mimic intercell hoppings $\omega e^{\pm ik}$, where the lattice momentum k is mapped to the phase shifts of phase shifters. Is the acoustic narrow tube essential for this setup? Could we entirely replace the acoustic structure with circuit-based controls to regulate the inter-cavity couplings? In the manuscript, a square-wave voltage is used for temporal modulation to realize the dynamic gain/loss, but such a non-smooth waveform may introduce high-frequency harmonics. How do these harmonics affect the PT symmetry of the system and the accuracy of momentum-band topology measurements? Would a sinusoidal modulation (with fewer harmonics) provide a more ideal platform for observing topological phases?

Version 1:

Reviewer comments:

Reviewer #1

(Remarks to the Author)

The authors replied to all of my comments, and I can recommend the publication of the manuscript in Nature Communications.

Reviewer #2

(Remarks to the Author)

The authors have nicely addressed all the concerns. Now it can be recommended to accept this paper on Nat. Comms.

Dear Reviewers,

We sincerely thank you for your time and insightful comments on our manuscript. Your feedback has been invaluable in enhancing both the quality and impact of our paper. Below, we provide a brief summary of the three review reports (in blue fonts) and our responses (in black fonts).

Reviewer #1 described our work as “a **fundamental result and the first direct and convincing experimental verification of the underlying bulk topology,**” and an “**important milestone in the research of time-dependent systems.**” Reviewer #3 praised our “**comprehensive experimental study of momentum-band topology**” and highlighted the “**direct bulk evidence for this emerging topological concept.**” Both reviewers emphasized the clarity and rigor of our work, and their feedback reflects strong support for the publication of our manuscript in *Nature Communications* after minor revisions.

Reviewer #2 raised some important concerns regarding the novelty and clarity of our contribution. We greatly appreciate Reviewer #2's careful reading of the manuscript and valuable suggestions. We respectfully acknowledge that these concerns may have arisen from an insufficient emphasis on the novelty and broader impact of our work in the original submission. In response, we have placed greater emphasis on the fact that our study not only provides **new theoretical insights** into momentum-band topology but also delivers **the first comprehensive, wavefunction-level experimental evidence** for this novel non-equilibrium topology, along with the innovative technique of reconstructing the Floquet operator.

To better highlight our contributions, we have extensively revised the manuscript, including, but not limited to, the following:

- The title has been updated to "Observation of Momentum-Band Topology in PT-Symmetric Floquet Lattices," which more effectively reflects the generality and broad applicability of our theoretical model.
- The Abstract, Introduction, and Conclusion have been substantially revised to clearly emphasize the novelty, technical innovation, and broader implications of our study.

Below, we provide detailed responses to all reviewers' comments and make corresponding revisions (**highlighted in yellow**) to enhance the manuscript's clarity and impact.

In conclusion, we propose a novel Floquet PT-symmetric model and present the first comprehensive experimental evidence for the emerging momentum-band topology. We believe that the revisions we have made successfully address all the review comments and significantly improve the manuscript. We respectfully request your reconsideration of the manuscript for publication in *Nature Communications*.

Best regards,

Prof. Chunyin Qiu

On behalf of all the authors

Response to Reviewer #1

The authors present an experimental demonstration of extracting the topological invariant of the momentum bandgap in time-modulated acoustic system. Momentum band-gaps occur in time-periodic systems in which the eigen-states form continuous bands that are gapped in the momentum. Unlike the more known energy bandgaps, here “forbidden” states (states in the gaps) do not conserve energy (draw or deplete energy from/to the modulation. Momentum band-gaps are relatively hard to realize (compared with energy gaps) especially in a direct way. As part of the increase in time-modulated system in recent years, several momentum band-gap phenomena were demonstrated, including the interface states in time that were also demonstrated here. Additionally, band topology is a well studied concept in energy band-gaps, but not so much for momentum band-gaps and were only proposed recently. Therefore, the observation presented here, of the topological invariant of a momentum gap is a fundamental result and the first direct and convincing experimental verification of the underlying bulk topology, and an important milestone at the research of time-dependent system. Additionally, the results are convincing and unambiguous, and the writing is pretty good. The additional result on the connection with the edge-states and non-Hermitian physics makes the work even richer. Therefore, I believe this manuscript is suitable to be published in Nature communications after minor revisions.

Response: We sincerely thank Reviewer #1 for the generous comments. We are honored by the recognition of our work as “a fundamental result” and “an important milestone at the research of time-dependent system.” These evaluations reinforce our belief in the broader significance of our contribution. Notably, the reviewer’s comment “the first direct and convincing experimental verification of the underlying bulk topology,” highlights the novelty of our experiments, and the remark “the results are convincing and unambiguous, and the writing is pretty good” further underscores the clarity and rigor of our work.

Finally, we are grateful for the strong support for publication in *Nature Communications*, pending minor revisions. We have carefully addressed all the helpful suggestions and provide detailed responses below.

1. in Fig.5b, regarding fact that the sound grows (and deviates from theory). This seems like the behavior seen in other works, in which the amplified mode that is the pair mode of the decaying mode that creates the edge-state start growing. If that is the case, I think the explanation can talk a little more about it – since this effect is seen in other works which showed only the edge-states (theory and experiment).

Response: We thank the reviewer for the thoughtful suggestion. As noted, the coexistence of both growing and decaying modes within the momentum bandgap leads to an inherent

sensitivity of the time-localized interface state to experimental imperfections. These imperfections, particularly in the system's parameter settings, result in the growth of the boundary states and the observed deviation from the theoretical predictions after $t \sim 25$ ms.

In response, we expand the discussion in the revised manuscript, with attention to previous works that observed similar behavior in edge states. Specifically, we have added the following clarification on page 11 (lines 245-250): “The deviation between the experimental and simulated results for $t > 25$ ms is primarily attributable to imperfections in system parameter settings⁴⁹. Specifically, the intensity pattern around the interface state is highly sensitive to the momentum (see Supplementary Information Note 5), since the Floquet lattices at both sides of the time interface support temporally growing in-gap modes, in addition to decaying modes akin to those in conventional spatial topological insulators^{21,34}.”

2. in Fig.2c the arrows in the sphere are too small and it's hard to understand it or see it. It would be good to make it clearer.

Response: We appreciate the reviewer's valuable suggestion. To enhance clarity, we have carefully enlarged the arrows in Fig. 2c and adjusted their contrast for better visibility.

3. Fig.2e – no x label.

Response: We thank the reviewer for catching this oversight. The missing x-axis label ' k/π ' has now been added to Fig. 2e, and we have verified the consistency of all axis labels throughout the manuscript.

4. Methods Details on Loss Compensation: In the Methods section, the authors mention a uniform background loss γ_0 and state that this can be compensated for. Could they clarify if the data presented in the main figures (e.g., Fig. 5) has been compensated? If so, briefly describing the procedure and perhaps showing a raw vs. compensated data plot in the Supplementary Information would enhance the reproducibility of the work.

Response: We appreciate this thoughtful suggestion. The data in all main figures have been compensated for background loss. To address this, we have included a brief description of the compensation procedure in the Methods section (see page 13, lines 329-333): “Note that the presence of uniform background loss γ_0 does not affect the essential physics but introduces an extra exponential decay $e^{2\pi\gamma_0 t}$ in the time-domain wavefunction. In this work, this decay is compensated by multiplying an exponential factor $e^{-2\pi\gamma_0 t}$ to the measured wavefunction (see Supplementary Information Note 6).” Meanwhile, to enhance the reproducibility of our work, we have provided the raw data in Supplementary Information

Note 6 (see page 11).

5. what is the meaning of virtual in virtual Dirac point? It is not clear to me. Maybe the authors can put a sentence on that.

Response: The term "virtual Dirac point" was used because, in the extreme case where the dynamic gain/loss parameter $\gamma = 0$, the system behaves as a static system without any modulation. In this scenario, the Dirac point arises due to the band folding caused by virtual Floquet modulation.

To avoid any potential confusion, we have replaced the term 'virtual Dirac point' with 'Floquet Dirac point' on page 4 (lines 81-85), accompanied by a brief but more precise explanation. The revised text now reads: "Clearly, it shows that the Floquet Dirac points at the EBZ boundaries, which are crossed by neighboring quasi-energy bands at $\gamma = 0$ because of the virtual Floquet band folding, split into pairwise Floquet EPs and form Floquet momentum gaps (featuring pairwise growing and decaying modes with complex-valued quasi-energies) due to the non-zero dynamic modulation, known as Floquet PT-symmetry breaking³⁸⁻⁴²."

6. Outlook on Causality in Temporal Pumping: The discussion of future work mentions that causality may lead to fundamentally distinct conclusions for temporal Thouless pumping. This is a very intriguing statement. Adding a single sentence to briefly speculate on what kind of new physics might be expected (e.g., directionality, quantization breakdown) would make the outlook even more exciting for the reader.

Response: We thank the reviewer for the insightful suggestion. In response, we have added the following sentence on page 11 (lines 266-270) to briefly speculate on what kind of new physics might be expected, which reads: "Looking ahead, the conclusive identification of momentum-band topology opens exciting directions for further experimental exploration, such as temporal Thouless pumping in lattice systems. The inherent causality constraint in temporal systems may give rise to phenomena fundamentally distinct from their spatial counterparts—for instance, the directionality arising from the one-way nature of time and the breakdown of quantization caused by non-Hermiticity."

Response to reviewer #2

In the manuscript titled by “Observation of momentum-band topology in acoustic PT-symmetric Floquet lattices”, authors focus on momentum band topology in the acoustic cavities. The static model is a PT symmetric two cavities which open $k=\pi$ momentum gap under PBC. Then, they added on-site time varying amplitude term in the type of alternative gain and loss, and the time-varying functions of two cavities are opposite to each one. They found that another momentum gap is opened away from $k=\pi$ which can support a temporal analogy of Jackiw rabbi mode-localized in a temporal interface. The experiment is based on two acoustic helmholtz resonator coupled by two tubes which is not tunable. The periodic boundary condition is realized by two phase shifters via electric circuit, and gain and loss are also given by electric circuit which act as additional in-phase and out-phase amplitude injection respectively.

Overall, the novelty of this paper is on the measurement of this acoustic 1D Floquet bands' winding number and states. However, considering the slim novelty and the following concerns, I cannot recommend its publication in Nat. com at this stage.

Response: We sincerely appreciate Reviewer #2's thoughtful evaluation of our manuscript. We acknowledge the reservations regarding the current suitability of our work for *Nature Communications*, which we believe primarily arise from our insufficient emphasis on the novel contributions of our study. We provide detailed clarifications below.

- **Scientific Novelty and Impact**—Momentum-band topology is a groundbreaking and rapidly emerging concept spanning diverse physics disciplines. Unlike the conventional energy-band topology defined over momentum space, it establishes a distinctive band topology in the energy Brillouin zone and predicts unique temporal topological interface modes (TIMs). Despite its potential to revolutionize the traditional paradigm of topological band theory, both the theoretical and experimental studies of this new concept remain in their infancy. Crucially, due to the inherent complexity of time-varying systems, momentum-band topology has, thus far, only been indirectly inferred through the observation of TIMs [Nat. Photon. 19, 518 (2025); Nat. Commun. 16, 707 (2025)].

In this work, for the first time, we theoretically establish the momentum-band topology in Floquet PT-symmetric lattices, and elucidate the intrinsic connection between fundamental PT-symmetric physics and this emerging band topology. (This theoretical contribution, though central to our work, was not adequately highlighted in our original manuscript.) Experimentally, by reconstructing Floquet operators and extracting eigenstates, we provide the first direct bulk evidence of momentum-band topology from both band-inversion and topological-invariant perspectives. This constitutes

decisive proof of the bulk topology [Note 1]. Furthermore, complemented by observations of temporal interface modes, our work establishes a comprehensive experimental framework for a deeper understanding of this novel band topology. Therefore, our study not only provides new theoretical insights into momentum-band topology but also delivers conclusive experimental evidence of this topology at the wavefunction level—a feat rarely achieved even in conventional topological physics.

[Note 1] Here we would like to highlight Reviewer #1's recognition of the fundamental nature of our work, specifically stating: "Therefore, the observation presented here, of the topological invariant of a momentum gap is a fundamental result and the first direct and convincing experimental verification of the underlying bulk topology, and an important milestone at the research of time-dependent system."

- **Technical and Methodological Advances**—To enable comprehensive, wavefunction-level characterization of momentum-band topology, we developed a novel technique to reconstruct the Floquet operator directly from measured time-domain wavefunctions. This fundamental operator, central to the theory of periodically driven systems, has long remained experimentally inaccessible [Note 2]. From the reconstructed Floquet operator, we extract the quasi-energies and eigenstates of the effective Hamiltonian. This enables direct experimental characterization of the complex-valued Floquet momentum gap and yields the first-ever measurement of a momentum-band topological invariant. This powerful approach addresses the long-standing challenge of experimentally probing the quasi-energy spectrum (even if complex-valued) and eigenstates of Floquet systems, thereby representing a significant technical step forward in temporal topological physics. Importantly, this foundational experimental technique can be widely applied in future studies of Floquet systems, including those involving more complicated dynamics, thereby enabling a comprehensive and deeper experimental understanding of periodically driven systems.

[Note 2] Our approach fundamentally differs from the method introduced in PRL 134, 256603 (2025), which is referred to in your Comment 5 below.

- **Broader Scientific Implications**—Our findings, based on a widely applicable tight-binding model, have broad interdisciplinary relevance, spanning condensed matter physics, integrated photonics, electrical circuits, and cold-atom systems. As a comprehensive demonstration of momentum-band topology, our work could inspire further experimental studies in temporal topological physics, such as novel temporal Thouless pumping in lattice systems (as noted by Reviewer #1). Additionally, our theoretical advance highlights the untapped potential of time-varying PT-symmetric lattices as a versatile platform for exploring temporal matter physics. This opens promising experimental avenues for investigating emerging concepts in time-varying

materials, including temporal quasicrystals, temporal Anderson localization, and temporal moiré superlattices—all of which were recently proposed in the context of photonic time crystals, but remain experimentally challenging due to fundamental limitations in optical modulation. Collectively, these innovative ideas could ultimately inform the design of next-generation functional devices with capabilities far beyond those of conventional Floquet systems.

In conclusion, our work introduces a novel dynamic PT-symmetric model for exploring momentum-band topology and presents the first comprehensive, wavefunction-level experimental evidence of this newly emerging non-equilibrium topological physics. This advance substantially enriches the field of temporal topology and marks a crucial step toward both understanding and applying these topological phases experimentally. We believe that the novelty, technical innovation, and broader implications of our study fully align with the high standards of *Nature Communications*. We would like to thank the reviewer again for the valuable feedback, which have significantly enhanced the quality and clarity of our manuscript.

Corresponding revisions made in our main text:

To more effectively reflect the generality and broad applicability of our theoretical model, we have revised the manuscript title to "*Observation of Momentum-Band Topology in PT-Symmetric Floquet Lattices*". This updated title moves beyond the specific acoustic implementation in favor of a more universal physical framework. Furthermore, to better highlight the novelty, technical innovation, and broad implications of our study, we have substantially revised the **Abstract**, **Introduction**, and **Conclusion paragraphs**. These sections now more clearly articulate our key contributions and the significance of our findings in the broader context of nonequilibrium topological physics.

- **Revised abstract (page 1).** Momentum-band topology is a groundbreaking concept in multidisciplinary physics. Unlike the conventional energy-band topology, it defines a distinctive band topology within the energy Brillouin zone. Despite revolutionizing the paradigm of topological band theory, both theoretical and experimental studies of this new concept remain in their infancy. Here, we unveil the momentum-band topology in a PT-symmetric Floquet lattice, where the drive-induced momentum gap can be rigorously characterized by a quantized Berry phase. Experimentally, we synthesize the model using an acoustic cavity-tube structure coupled to custom-designed external circuits. By innovatively reconstructing the Floquet operator from measured time-domain wavefunctions, we extract the system's eigenstates and, for the first time, provide direct bulk evidence of momentum-band topology via momentum-band inversion and topological invariants. This is accompanied by a clear observation of time-localized interface states, thus providing a comprehensive examination of the

temporal bulk-boundary correspondence. Our work paves the way for further experimental studies on the burgeoning momentum-gap physics.

- **Revisions made in the introduction (page 3).** Despite its revolutionary paradigm, the full theoretical elaboration of this new concept remains in its early stages. Furthermore, while experimental breakthroughs have been recently achieved in spatially discrete systems^{34,35,37}, the explorations are currently confined to the observation of TIMs, which provide only indirect evidence of momentum-band topology. Crucially, direct evidence of the bulk topology—the experimental characterization of the topological invariant itself—remains out of reach, hindered by the inherent complexity of time-varying systems and substantial technical challenges in extracting eigenstates. These critical bottlenecks underscore the pressing need for both theoretical transformation and experimental innovation.

In this work, we theoretically propose and experimentally demonstrate momentum-band topology in a novel PT-symmetric Floquet lattice. This model, which is broadly relevant to various experimental platforms, exhibits a nontrivial momentum gap induced by the periodic modulation of onsite gain and loss. The resulting momentum-band topology can be rigorously characterized by a quantized Berry phase defined over the EBZ.

- **Updated Conclusion and Discussion paragraphs.** We theoretically unveil the momentum-band topology in Floquet PT-symmetric lattices and elucidate its intrinsic connection with fundamental PT-symmetric physics. Built upon a widely applicable tight-binding model, our findings exhibit broad interdisciplinary relevance, spanning condensed matter physics, integrated photonics, electric circuits, and cold-atom systems. Furthermore, we present a comprehensive, wavefunction-level experimental study of the momentum-band topology using acoustic Floquet lattices. We not only provide the first direct bulk evidence of momentum-band topology via band-inversion and topological invariants, but also demonstrate the intriguing TIM in real physical time, thus complementarily highlighting the role of temporal bulk-boundary correspondence.

Looking ahead, the conclusive identification of momentum-band topology opens exciting directions for further experimental exploration, such as temporal Thouless pumping in lattice systems. The inherent causality constraint in temporal systems may give rise to phenomena fundamentally distinct from their spatial counterparts—for instance, the directionality arising from the one-way nature of time and the breakdown of quantization caused by non-Hermiticity. In addition, the key experimental technique developed here—reconstructing the Floquet operator via measuring the temporal evolution of wavefunctions—effectively addresses the long-standing challenge of probing the complex-valued quasi-energy spectrum and eigenstates of non-Hermitian Floquet systems. This method holds broad applicability for future studies of Floquet

non-equilibrium systems, enabling comprehensive, wavefunction-level experimental access to periodically driven quantum dynamics. Last but not least, our theoretical and experimental results demonstrate significant potential for exploring a wide range of temporal matter physics in dynamic PT-symmetric systems through the control of time-dependent gain and loss. This includes, for example, temporal quasicrystals⁵⁷, time-domain Anderson effects^{58,59}, and temporal moiré superlattices^{60,61}. These novel concepts could enable the design of functional devices with capabilities that go beyond conventional Floquet systems.

1 Clarification Required on Gap Opening:

Does the opening of the k-gap (not $k=\pi$ one) require the static gain and loss? The temporal interface mode is attributed to Floquet SSH temporal domain model, as a analogy of Jackiw-Rebbi zero modes (from their formula). The static gain and loss gives $k=\pi$ gap, which exhibits large amplitude exponential growth factor.

Response: We thank the reviewer for this insightful question and thoughtful comments. To clarify, the formation of the Floquet k-gap at $k \neq \pi$ does **not** require static gain and loss. As shown in Fig. R1, the Floquet k-gap remains open even in the absence of the static gain and loss ($\gamma_s = 0$, left panel).

Figure R1. Band structures for the Floquet lattices without (left) and with (right) applying static gain/loss, where the dark and grey lines represent the real and imaginary parts of the energy, respectively. The green band highlights the isolated momentum band with nontrivial topology in energy Brillouin zone.

While static gain and loss are not essential for opening the Floquet k-gap, their introduction (which opens the *static* k-gap at $k = \pi$) serves two primary purposes:

- It creates well-defined, isolated momentum bands to facilitate the characterization of topological invariants in the EBZ. In contrast, in the absence of static gain and loss

($\gamma_s = 0$), the degeneracy at $k = \pi$ makes the definition of topological invariants more complicated.

- It enables a direct comparison between the topological properties of the static k-gap and those of the Floquet k-gap, as demonstrated in Fig. 2e and Fig. 5. This comparison highlights the unique momentum-gap topology inherent to our PT-symmetric *Floquet* lattice.

We hope this answers the reviewer's question. If further clarification is needed regarding the role of static gain and loss or the analogy to Jackiw-Rebbi modes, we are happy to discuss it further. In response, we have added the following sentences in our revised manuscript (page 5, lines 87-90): "The introduction of the static momentum gap serves two primary purposes: it will facilitate the characterization of topological invariants in the EBZ and enable a direct comparison to the Floquet momentum gap with nontrivial topology. (This comparison highlights the unique momentum-gap topology inherent to our PT-symmetric Floquet lattice.)"

2 Concerns Regarding "Time Modulation" Experimental Implementation:

It is hard to say the time varying in this manuscript is genuine time modulation. The mechanism of this experiment is based on fast electric sensor for acoustic amplitude, fast operation (compare with acoustic frequency), and injection (the ending of this loop). It is a time-periodic energy cancellation or injection, instead of modulating some physical parameters. On this basis, it makes little sense for "a broad range of temporal matter physics. This includes, for example, temporal quasicrystals, time-domain Anderson effects, and temporal Moiré superlattices".

Response: We appreciate the reviewer's insightful comments regarding the physical essence of "time modulation" and the consequent broader implication of our work. We address these points as follows:

- **On "whether time modulation constitutes genuine modulation of physical parameters"**

We fully agree with the reviewer that genuine time modulation should act on the intrinsic physical parameters of a system. In this study, we actively modulate acoustic gain and loss, which, in the non-Hermitian lattice model, correspond to the imaginary part of the on-site energy. In previous works [48,52], acoustic gain and loss in *static systems* have been successfully realized by applying in-phase and out-of-phase acoustic feedback to individual cavities (which emulate lattice sites). The workflow can be summarized as follows: the sound signal within a given cavity is picked up by a detector, modulated by a preamplifier and a phase shifter, and finally fed back to the cavity through a speaker.

To achieve time-periodic gain and loss, we augment the system with additional control units. A wave generator temporally modulates the amplification factor of a voltage-controlled amplifier (VCA), enabling dynamic tuning of gain and loss beyond the static configurations demonstrated in previous studies [Note 3]. Therefore, our experimental setup indeed realizes *genuine* temporal modulation of the fundamental physical parameters (i.e., acoustic gain and loss), which moves beyond a simple energy injection/cancellation loop [Note 4].

[Note 3] The voltage signal generated by the wave generator does not directly enter the gain/loss circuit. Instead, it only modulates the strength of the gain/loss by adjusting the amplification factor of the VCA. To avoid any misunderstandings, we have clarified this in the revised manuscript (page 12, lines 304-307): “Note that the voltage output from the wave generator does not directly participate in the energy exchange of the gain/loss circuits, but rather dynamically switches the amplification factor of the VCA, thereby realizing the time-dependent amplitude modulation of gain and loss.”

[Note 4] Note that although the acoustic gain and loss are periodically modulated, the sound energy itself does not exhibit periodic variation. This is clearly demonstrated in Fig. 5. For instance, the sound energy at $k = \pi$ grows exponentially over time. Inspired by your comment, we have added the following sentence on page 12 (lines 307-309): “The periodic modulation of gain and loss does not result in a simple cycle of sound energy injection and cancellation (see, e.g., Figs. 5c and 5d). This behavior reflects the dynamic nature of the underlying physical parameters.”

➤ **On "relevance to other temporal matter physics"**

Our work has successfully demonstrated unique momentum-gap topological physics via periodic gain/loss modulations, which experimentally validates the effectiveness of dynamically modulating physical parameters in our acoustic platform. (Similarly, one can achieve time-modulated acoustic hoppings, the other fundamental physical parameter in non-Hermitian systems.) Building on this, we can further explore a wide range of emerging temporal matter physics. For instance, one could study temporal quasicrystals, time-domain Anderson effects, and temporal Moiré superlattices by applying quasi-periodic, disordered, and Moiré-type temporal modulations to the gain/loss or hopping configurations, respectively. In light of these insights, we have revised a conclusion sentence to the following (page 11, lines 276-281): “Last but not least, our theoretical and experimental results demonstrate significant potential for exploring a wide range of temporal matter physics in dynamic PT-symmetric systems through the control of time-dependent gain and loss. This includes, for example, temporal quasicrystals⁵⁷, time-domain Anderson effects^{58,59}, and temporal moiré superlattices^{60,61}. These novel concepts could enable the design of functional devices with capabilities that go beyond conventional Floquet systems.”

3 Ambiguities in Temporal Interface State Observations (Fig. 5 and Fig. S4):

The temporal localization is only at a specified k at the k -gap around 0.2 - 0.5π . Yet, a small deviation from that value makes the intensity exponentially grow. How such exponentially growing modes exist? From the spectrum, it should be evanescent one as in the k gap there should be only interface states at the specified k realized via phase shifters in circuits. For such acoustic experiment with fast electric device control, such deviation should be clarified. Also, the increasing of the tail seems not localized when observing it beyond 40 ms.

Response: We thank the reviewer for the helpful feedback. As pointed out correctly by the reviewer, theoretically, there is only one in-gap state (with real energy) that decays on both sides of the time interface at a specific k , akin to the well-known spatial TIMs. However, in contrast to the conventional energy gap that supports only spatially evanescent modes, the momentum gap here hosts pairwise modes that grow and decay with time [Note 5], characterized by eigen-energies with the same real parts but opposite imaginary components (see Figs. 2b and 4a) [21,33-37]. The presence of growing modes (for all in-gap momenta) makes the observation of temporal TIMs more challenging than their spatial counterparts. Any slight momentum deviation can excite the growing modes and destroy the perfect decay pattern away from the time interface (see Fig. S4 supplied below). This phenomenon is consistent with the recent experimental observation of temporal TIMs in photonic discrete-time systems (see Ref. [34]). In our setup, the imperfections in the system's parameter settings (e.g. the finite resolution of phase shifter, $\sim 0.03\pi$, which effectively induces a momentum shift) lead to the deviation between our experimental and simulated results. Nevertheless, our experimental results clearly distinguish between topologically nontrivial and trivial temporal interfaces, as shown in Figs. 5b and 5d.

[Note 5] As detailed in our response to Comment 6, these modes are excluded from the momentum spectrum in Fig. 2e by imposing time-periodic boundary conditions.

Supplementary Fig. S4. Sound intensity evolutions simulated for the time-domain wall systems at

momenta $k = k_D + \delta k$, with $\delta k = 0$, $\delta k = 0.02\pi$, and $\delta k = 0.08\pi$. The results illustrate a high momentum sensitivity of the acoustic time-localized interface state.

In summary, the observed deviations arise from both the coexistence of in-gap growing modes and imperfections in the system parameter settings. In particular, the presence of growing modes reflects a fundamental difference from conventional topological systems that host spatial TIMs.

To clarify this point, we have provided a detailed discussion in the revised manuscript (page 11, lines 245-250), which reads: "The deviation between the experimental and simulated results for $t > 25$ ms is primarily attributable to imperfections in system parameter settings⁴⁹. Note that the intensity pattern around the interface state is highly sensitive to the momentum (see Supplementary Information Note 5), since the Floquet lattices at both sides of the time interface support temporally growing in-gap modes, in addition to decaying modes akin to those in conventional spatial topological insulators^{21,34}."

4 Insufficient Resolution near Exceptional Points (EP)

In the other EP experimental observations, the relative low resolution around EP is mostly due to the parameter shift or disorder. Yet for the experiment here, as all the couplings, periodic boundary condition k , time varying, gain and loss in acoustic cavities are precisely controlled by electric device- which is quite accurate and fast enough for varying the acoustic amplitude, acoustic amplitude detection & injections. Why there is still low resolution around EP in the results shown in Fig. 4?

Response: We thank the reviewer for this thoughtful question. Like other EP experimental observations, our experiment setup is also subject to undesired errors in system parameters. Specifically, fabrication imperfections, temperature fluctuations, and environmental noise introduce unavoidable errors in both the cavity-tube and circuit systems (a detailed discussion on circuit errors can be found in Ref. [49]). These imperfections reduce the resolution near the EP. Nevertheless, the precision of our experiment is sufficient to unveil the presence of the momentum gap, especially the Floquet gap around $k = 0.3\pi$, which is central to this work and has not been demonstrated before. As clearly shown in Fig. 4a (right panel), the spectrum around the Floquet momentum gap exhibits nearly identical real parts of the energy but opposite imaginary parts—providing direct evidence of the dynamic momentum gap.

To address this point, we have added a sentence on page 8 (lines 184-186): "The deviation between the experimental and theoretical results arises from noise and imperfections in system parameters⁴⁹, e.g., the hopping w , and the gain/loss parameters γ_s and γ ."

5 Insufficient Literature Coverage:

For example, The model without static gain and loss has been revealed and experimentally checked in “Liu, Weijie, et al. "Floquet parity-time symmetry in integrated photonics." *Nature Communications* 15.1 (2024): 946.”. In addition, It is nice idea to use two different input to rebuild the Floquet operator. This is a kind of tomographic of Floquet operator. Yet, the prior literature is not properly mentioned. I recommend author refer to Asapanna, Rajesh, et al. "Observation of extrinsic topological phases in Floquet photonic lattices." *Physical Review Letters* 134.25 (2025): 256603.” Also, for the realization of periodic boundary condition, it has been proposed and realized experimentally in both acoustic and microwave platforms. Please refer to: Chen, Zhao-Xian, et al. "Direct measurement of topological invariants through temporal adiabatic evolution of bulk states in the synthetic Brillouin zone." *Physical Review Letters* 134.13 (2025): 136601. Zhang, Zhe, Pierre Delplace, and Romain Fleury. "Anomalous topological waves in strongly amorphous scattering networks." *Science Advances* 9.12 (2023): eadg3186.

Response: We thank the reviewer for pointing out these valuable references and for the positive feedback on our experimental technique (regarding the use of two different inputs to rebuild the Floquet operator). We have carefully reviewed these works and cited them appropriately in our revised manuscript:

[41] Liu, W., et al. Floquet parity-time symmetry in integrated photonics. *Nat. Commun.* 15, 946 (2024).

[51] Zhang, Z., Delplace, P., & Fleury, R. Anomalous topological waves in strongly amorphous scattering networks. *Sci. Adv.* 9, eadg3186 (2023).

[52] Chen, Z. et al. Direct Measurement of Topological Invariants through Temporal Adiabatic Evolution of Bulk States in the Synthetic Brillouin Zone. *Phys. Rev. Lett.* 134, 136601 (2025).

[56] Asapanna, R. et al. Observation of Extrinsic Topological Phases in Floquet Photonic Lattices. *Phys. Rev. Lett.* 134, 256603 (2025).

We would like to respectfully note that, while Refs. [41] and [56] share thematic connections with our study, they do not detract from the novelty or the primary contributions of our work. Specifically:

- Ref. [41] explores Floquet PT-symmetry breaking in a two-level integrated photonics system, with a primary focus on the emergence of growing modes. While this reference is related to our work, it does not cover the core aspects of our research—momentum-band topology and temporal TIMs.
- The recent work [56] experimentally explores extrinsic topological phases in Hermitian Floquet photonic lattices. To extract eigen-energies and eigenvectors, the authors measure the time evolution of excited states and perform a space-time Fourier transform to the data. The energy-momentum spectrum is achieved from the peak values of the two sublattice Fourier spectra, while eigenvectors are inferred from the

associated amplitude and phase information. However, this method cannot access complex-valued spectra of non-Hermitian systems and can only provide qualitative signatures of momentum-gap growing modes via enhanced spectral density. In contrast, we reconstruct the Floquet operator and the effective Hamiltonian from the measured time-domain wavefunctions. This enables direct measurement of the complex-valued quasi-energies and eigenstates in non-Hermitian Floquet lattices. This, in turn, facilitates our comprehensive, wavefunction-level characterization of the momentum band topology. In particular, by requiring only the time-domain wavefunction within a single Floquet period, our approach remains simple yet effective (even in the presence of considerable background loss). To the best of our knowledge, it represents the first experimental scheme for reconstructing Floquet operators, which holds broad applicability for future explorations of Hermitian or non-Hermitian Floquet systems.

To clarify the distinction from Ref. [56], we have included the following statement on pages 7 and 8 (lines 174-177): “Our experimental scheme, based on the reconstructed Floquet operator, enables direct measurement of complex-valued energy spectra in non-Hermitian Floquet lattices. This technique significantly differs from previous ones, where only real-valued band structures are extracted from space-time Fourier spectra⁵⁴⁻⁵⁶.” Furthermore, to underscore the potential impact and implications of this novel experimental technique, we have added these sentences to the Conclusion (page 11, lines 271-276): “In addition, the key experimental technique developed here—reconstructing the Floquet operator via measuring the temporal evolution of wavefunctions—effectively addresses the long-standing challenge of probing the complex-valued quasi-energy spectrum and eigenstates of non-Hermitian Floquet systems. This method holds broad applicability for future studies of Floquet non-equilibrium systems, enabling comprehensive, wavefunction-level experimental access to the dynamics of periodically driven systems.”

We thank the reviewer again for bringing these interesting references to our attention, which have helped improve the manuscript.

6 Incomplete Information for Fig. 2e and Fig. 2f:

The real-space configuration for the state depicted in Fig. 2f and the imaginary component in Fig. 2e are not described. This information is crucial for proper interpretation and understanding of the mode structure.

Response: We thank the reviewer for pointing out this important issue, which is indeed important for the proper interpretation and understanding of the mode structure.

➤ **Regarding the imaginary component of the eigenenergy in Fig. 2e.**

The eigenenergy corresponding to the momentum spectrum in Fig. 2e has no imaginary components, as explained below. Technically, to solve for the momentum spectrum (Fig. 2e) and eigenstates (Fig. 2f) of the temporal domain-wall structure, we apply the temporal periodic boundary condition $\psi_{t=NT} = \psi_{t=-NT}$, and the spatial Bloch boundary condition $\psi_{x=m+1} = e^{ik}\psi_{x=m}$, where $2N = 60$ characterizes temporal duration of the time-domain wall structure and m indices the spatial position of unit cell. Given the connection between $\psi_{t=NT}$ and $\psi_{t=-NT}$ via the time evolution operator $\mathbf{U}_{\text{DW}}(k)$ of the temporal domain-wall structure, i.e., $\psi_{t=NT} = \mathbf{U}_{\text{DW}}(k)\psi_{t=-NT}$, the temporal periodic boundary condition $\psi_{t=NT} = \psi_{t=-NT}$ enforces that the eigenvalue of $\mathbf{U}_{\text{DW}}(k)$ must be 1. Using the definition of quasi-energy, $e^{-i2NT\varepsilon}\psi = \mathbf{U}_{\text{DW}}\psi$, we find that all eigenstates in Fig. 2e correspond to quasi-energies $\varepsilon = 0$. As such, all eigenstates have zero imaginary components in their eigen-energies. In other words, only the real-valued TIMs emerge within the momentum gap, while the in-gap growing and decaying modes are excluded from the momentum spectrum of the temporal domain-wall structure.

To clarify this point, we have added the following sentence to our revised manuscript (page 5, lines 109-112): “Note that owing to the implementation of the time-periodic boundary condition, all eigenstates have zero imaginary components in their eigenenergy, and the in-gap growing and decaying modes are excluded from the momentum spectrum.” Additionally, we have rewritten the corresponding paragraph in Supplementary Information Note 3 (see page 7) for further details.

➤ **Regarding the real-space configuration for the state depicted in Fig. 2f.**

Figure 2f presents the probability density distributions $|\psi|^2$ of both the bulk mode and the TIM across the temporal domain-wall structure, which are spatially *uniform* under the spatial Bloch boundary condition. Therefore, Fig. 2f shows only the temporal pattern of the wavefunctions. To make this point clearer, we have added the following sentence on page 5 (lines 115-116): “Note that the probability density distribution of the states is spatially uniform due to the implementation of spatial Bloch boundary condition.”

7 Clarification of "Virtual Dirac Point":

The concept and implications of a "virtual Dirac point" introduced on page 5 (first line) remain unclear. This discussion must be refined for better readability and scientific clarity.

Response: Thank you for pointing out this ambiguity. The term "virtual Dirac point" was used to describe a scenario where, in the extreme case of $\gamma = 0$ (no dynamic gain/loss), the

system behaves essentially as a static system without modulation. Therefore, the Dirac points arise from the virtual folding of the Floquet bands, which is physically imperceptible.

To avoid potential confusion and better reflect the physical scenario involved, we have replaced 'virtual Dirac point' with 'Floquet Dirac point' and explained the physical origin behind this concept. The revised sentence now reads (on page 4, lines 81-85): “Clearly, it shows that the Floquet Dirac points at the EBZ boundaries, which are crossed by neighboring quasi-energy bands at $\gamma = 0$ because of the virtual Floquet band folding, split into pairwise Floquet EPs and form Floquet momentum gaps (featuring pairwise growing and decaying modes with complex-valued quasi-energies) due to the non-zero dynamic modulation, known as Floquet PT-symmetry breaking³⁸⁻⁴².”

8 Figure Improvement:

The azimuthal axis in Fig. 2e should be explicitly labeled for better clarity.

Response: We appreciate the reviewer for pointing this out. We have added the azimuthal axis ' k/π ' to Fig. 2e.

Response to Reviewer #3

This manuscript presented a comprehensive experimental study of momentum-band topology in a PT-symmetric Floquet lattice, addressing the lack of direct bulk evidence for this emerging topological concept. Using an acoustic cavity-tube system coupled with active feedback circuits, the researchers synthesized a Floquet lattice where periodic gain/loss modulations induced momentum gaps. By reconstructing the effective Hamiltonian from time-evolved wavefunctions, they experimentally confirmed momentum-band inversion and quantified the topological invariant in the energy Brillouin zone, providing the evidence of momentum-band topology. Additionally, they observed temporally localized interface states at a topological domain wall, verifying the temporal bulk-boundary correspondence. The manuscript presents an interesting experimental exploration of momentum-band topology in acoustic systems, and it is well-written. However, I still have a few more questions and suggestions:

Response: We sincerely thank Reviewer #3 for the thoughtful and comprehensive review of our manuscript. We greatly appreciate the positive assessment of the quality of our work, as well as the constructive suggestions provided. We have carefully considered all the reviewer's questions and suggestions, and address them point by point below.

1. Please check whether the x-axis label is missing in Fig. 2e and Fig. 4b.

Response: We thank the reviewer for bringing this to our attention. The missing x-axis labels have now been added to Fig. 2e and Fig. 4b in our revised manuscript.

2. The manuscript states that the narrow tube provides a static and reciprocal intracell coupling ($\omega=36$ Hz), while also introducing two sets of unidirectional coupling circuits to mimic intercell hoppings ($\omega e^{\pm ik}$), where the lattice momentum k is mapped to the phase shifts of phase shifters. Is the acoustic narrow tube essential for this setup? Could we entirely replace the acoustic structure with circuit-based controls to regulate the inter-cavity couplings?

Response: We thank the reviewer for the insightful questions. The acoustic narrow tube is not essential for our setup. It is indeed feasible to replace the acoustic narrow tubes with coupling circuits to achieve intracell coupling. In this work, we opted to use acoustic tubes for intracell coupling to simplify the experimental setup, as they offer a straightforward and reliable means of realizing static coupling. However, we agree that in scenarios where tunable intracell coupling is required, circuit-based coupling would be a more suitable alternative.

3. In the manuscript, a square-wave voltage is used for temporal modulation to realize the dynamic gain/loss, but such a non-smooth waveform may introduce high-frequency harmonics. How do these harmonics affect the PT symmetry of the system and the accuracy of momentum-band topology measurements? Would a sinusoidal modulation (with fewer harmonics) provide a more ideal platform for observing topological phases?

Response: We thank the reviewer for arising this thoughtful comment. In our study, the square-wave modulation $S(t) = \text{sgn}[\cos(2\pi\Omega t)]$ is employed both theoretically and experimentally. We note that all Fourier harmonics $S_n(t) = s_n \cos(2\pi n\Omega t)$ of this waveform preserve even symmetry with respect to the center of the temporal unit cell. As a result, the Floquet operator inherits the PT-symmetry from the instantaneous Hamiltonian [see Phys. Rev. B 82, 235114 (2010)]. In our experiment, the modulation speed of the circuit is much faster than the dynamics of sound waves, thereby enabling accurate implementation of both square-wave and sinusoidal modulations. However, we agree with the reviewer that in cases involving slower modulation speeds, where accurately characterizing higher harmonics of square waves becomes challenging, a smooth sinusoidal modulation would indeed offer a more ideal platform for observing topological phases. The choice of square-wave modulation in our experiment was primarily motivated by practical considerations: it is simpler to implement, as it only requires periodic switching of the voltage-controlled amplifier (VCA), which is straightforward and efficient for our setup.

Response Letter

We really appreciate the valuable comments given by all the reviewers on this manuscript. Below, we provide detailed responses to all reviewers' comments.

Response to Reviewer #1

The authors replied to all of my comments, and I can recommend the publication of the manuscript in Nature Communications.

Response: We thank Reviewer #1 for recommending the publication of our work in Nature Communications.

Response to Reviewer #2

The authors have nicely addressed all the concerns. Now it can be recommended to accept this paper on Nat. Comms.

Response: We thank Reviewer #2 for recommending the acceptance of this paper on Nature Communications.